# Infection with chikungunya virus confers heterotypic cross-neutralizing antibodies and memory B-cells against other arthritogenic alphaviruses predominantly through the B domain of the E2 glycoprotein

John M. Powers[1☯¤a], Zoe L. Lyski[2☯¤b], Whitney C. Weber[1,2☯], Michael Denton[1], Magdalene M. Streblow[1], Adam T. Mayo[1], Nicole N. Haese[1], Chad D. Nix[2], Rachel Rodríguez-Santiago[3], Luisa I. Alvarado[3], Vanessa Rivera-Amill[3], William B. Messer[2,4,5]*, Daniel N. Streblow[1,6]*

1 Vaccine and Gene Therapy Institute, Oregon Health and Science University, Beaverton, Oregon, United States of America, 2 Department of Molecular Microbiology and Immunology, Oregon Health and Science University, Portland, Oregon, United States of America, 3 Ponce Health Sciences University/ Ponce Research Institute, Ponce, Puerto Rico, 4 Department of Medicine, Division of Infectious Disease Oregon Health and Science University, Portland, Oregon, United States of America, 5 OHSU-PSU School of Public Health, Program in Epidemiology, Oregon Health and Science University, Portland, Oregon, United States of America, 6 Division of Pathobiology and Immunology, Oregon National Primate Research Center, Beaverton, Oregon, United States of America

☯ These authors contributed equally to this work.
¤a Current address: Department of Epidemiology, University of North Carolina at Chapel Hill, Chapel Hill, North Carolina, United States of America
¤b Current address: Department of Immunobiology, University of Arizona, Tucson, Arizona, United States of America
* messer@ohsu.edu (WBM); streblow@ohsu.edu (DNS)

## Abstract

Infections with Chikungunya virus, a mosquito-borne alphavirus, cause an acute febrile syndrome often followed by chronic arthritis that persists for months to years post-infection. Neutralizing antibodies are the primary immune correlate of protection elicited by infection, and the major goal of vaccinations in development. Using convalescent blood samples collected from both endemic and non-endemic human subjects at multiple timepoints following suspected or confirmed chikungunya infection, we identified antibodies with broad neutralizing properties against other alphaviruses within the Semliki Forest complex. Cross-neutralization generally did not extend to the Venezuelan Equine Encephalitis virus (VEEV) complex, although some subjects had low levels of VEEV-neutralizing antibodies. This suggests that broadly neutralizing antibodies elicited following natural infection are largely complex restricted. In addition to serology, we also performed memory B-cell analysis, finding chikungunya-specific memory B-cells in all subjects in this study as remotely as 24 years post-infection. We functionally assessed the ability of memory B-cell derived antibodies to bind to chikungunya virus, and related Mayaro virus, as well as the highly conserved B domain of the E2 glycoprotein thought to contribute to cross-reactivity between related Old-World alphaviruses. To specifically assess the role of the E2 B domain in cross-

**Data Availability Statement:** All relevant data are within the manuscript and its Supporting Information files.

**Funding:** The work presented in this manuscript was supported by grants from the National Institutes of Health 1U19AI142790 (DNS), R01AI153434 (WBM), R21AI135537(WBM), UL1TR002369 (WBM), Takeda IISR 2016-101586 (WBM), and the Sunlin and Priscilla Chou foundation (WBM). Centers for Disease Control and Prevention U01CK000437 (VRA) and U01CK000580 (VRA) and T32GM142619 (WCW). The funders had no role in study design, data collection and analysis, decision to publish, or preparation of this manuscript.

**Competing interests:** The authors have declared that no competing interests exist.

neutralization, we depleted Mayaro and Chikungunya virus E2 B domain specific antibodies from convalescent sera, finding E2B depletion significantly decreases Mayaro virus specific cross-neutralizing antibody titers with no significant effect on chikungunya virus neutralization, indicating that the E2 B domain is a key target of cross-neutralizing and potentially cross-protective neutralizing antibodies.

## Author summary

The emergence and re-emergence of alphaviruses as important human pathogens raises questions about the durability and breadth of alphavirus immunity following natural infection in humans. In this study, we examine human immune sera from twelve individuals infected (up to 24 years) previously with chikungunya virus and test the sera against a panel of five Old-World arthritogenic alphaviruses and one New-World encephalitic alphavirus. Both homotypic and cross-reactive memory B-cells were identified in subjects out to 24 years post infection. Our results indicate that infection with chikungunya virus results in a robust and durable cross-reactive humoral immune response. Such a response could potentially provide immunity against repeat infection with chikungunya as well as related alphaviruses for years to decades after initial infection. This cross-reactivity might contribute to restricted transmission of closely related alphaviruses and indicates the potential for chikungunya candidate vaccines to elicit broad protection against other alphaviruses in the Semliki Forest complex.

## Introduction

Alphaviruses, members of the family *Togaviridae*, are a large group of arthropod-borne viruses with worldwide distribution that cause both sporadic outbreaks and epidemics. These predominantly mosquito-borne viruses have a wide host range and can replicate in a variety of cell types [1–3]. Alphaviruses are broadly grouped in seven distinct antigenic complexes–Barmah Forest, Eastern Equine Encephalitis, Middleburg, Ndumu, Semliki Forest, Venezuelan Equine Encephalitis, and Western Equine Encephalitis [4]. These viruses can be broadly divided into two categories, New and Old World, based on phylogenetic relatedness and clinical manifestations of disease. While infections with Old World alphaviruses, such as chikungunya virus (CHIKV) and Mayaro virus (MAYV) predominantly cause myalgia and arthralgia, New World alphaviruses such as Venezuelan equine encephalitis virus (VEEV) and Eastern equine encephalitis virus (EEEV) infections can cause life-threatening encephalitis.

Of the alphavirus members, CHIKV has the widest global distribution, with CHIKV transmission reported in over 100 countries worldwide [5,6]. Before 2013 CHIKV had not yet been locally acquired or transmitted within the Americas [7]. Historically, circulating predominantly in regions of Africa and Asia, CHIKV emerged on a global scale in the mid-2000s, resulting in outbreaks in Africa, Asia, as well as the Caribbean and North, Central, and South Americas, leading to almost 2 million reported infections [8]. At the time of the writing of this manuscript (October 27th 2022), there have been 338,592 cases with 70 deaths in 2022, with the majority of the cases occurring in Brazil (ECDC). In the current investigation, we characterize samples from an endemic human cohort in Puerto Rico. The island of Puerto Rico experienced a CHIKV epidemic starting in May 2014 with official surveillance reporting 28,327 suspected cases and 31 deaths by the epidemic's end [9].

Other related Old-World alphaviruses include O'nyong nyong virus (ONNV), which forms a monophyletic group with CHIKV. ONNV is endemic in sub-Saharan Africa and periodically causes outbreaks in West and East Africa [10,11]. Mayaro (MAYV) and Una (UNAV) viruses are closely related alphaviruses that commonly cause disease outbreaks in Central and South America [12]. The most distant member of the SFV complex that we included in our alphavirus panel is Ross River virus (RRV), which is endemic to Australia and several neighboring Pacific Islands [13]. Outside of the SFV complex are the distantly related New World encephalitic alphaviruses that circulate in North, South, and Central America.

In general, alphaviruses are ~70 nm enveloped viruses with an icosahedral capsid of $T = 4$ symmetry that is composed of 240 capsid monomers. Each virus particle contains ~10–12 kb single-stranded, positive sense RNA genome that contains two open reading frames, both translated with a 5' cap and 3' poly-A tail [13–15]. The viral genome encodes four nonstructural proteins (nsP1 –nsP4) involved in RNA replication, and five structural proteins (Capsid, E3, E2, 6K, E1) required for viral encapsidation and budding [2,16,17]. Structural E1-E2 heterodimers trimerize to form the surface spikes of the virus envelope responsible for attachment and entry into host cells. Specifically, E2 is responsible for cellular receptor binding, and E1 mediates membrane fusion [17]. The structural proteins E1 and E2 are key targets of the host antibody response. In humans and mice, the antibody response is primarily generated against E2 [18–21]. Previous studies have reported the development of cross-neutralizing antibodies (Abs) in model organisms and humans following infection with SFV complex alphavirus members, and the B domain of the E2 (E2 B) glycoprotein has been implicated as a potential target for broadly cross-neutralizing antibodies due to the disruption of the trimeric spike [21–24].

Virus-specific Abs are initially secreted by short-lived plasma cells to help combat the current infection. Virus-specific B-cells further differentiate in germinal centers of peripheral lymph nodes where they undergo affinity maturation and exit the lymph node as one of two types of long-lived memory cells. One cell type, long-lived plasma cells (LLPCs), traffic to bone marrow where they secrete large amounts of antigen-specific Abs that circulate in the serum for months to years post-exposure [25]. LLPC-derived Abs are thought to protect against repeat infections with homologous or closely related pathogens and are often regarded as the first line of defense. Memory B-cells (MBCs) also differentiate in germinal centers and circulate in low numbers in peripheral blood. MBCs do not secrete Abs, but instead patrol peripheral circulation for invading pathogens, poised to quickly respond to repeat infections by proliferating and differentiating into Ab secreting cells. It has been reported that MBCs respond to related but antigenically distinct pathogens that evade preexisting serum Abs [26,27]. Consequently, MBCs have the potential to play a critical role in developing broad immunity especially in the face of waning Ab titers and the emergence of new closely related alphaviruses.

To further characterize the durability and breadth of cross-reactive anti-alphavirus Abs and MBCs, we evaluated a panel of convalescent samples from subjects enrolled in one of two larger human arbovirus cohorts. The first, a non-endemic (travelers) cohort based in Portland, Oregon and the second, an endemic cohort based in Ponce, Puerto Rico. Subjects had suspected or confirmed CHIKV infection, further confirmed by serology (CHIKV 50% neutralization titer > 1:20) and samples from three alphavirus naïve subjects (CHIKV 50% neutralization titer < 1:20) were included as controls (Table 1). We evaluated study participants for the presence of CHIKV neutralizing antibodies, cross-alphavirus neutralizing antibodies, and CHIKV-specific and cross-reactive memory B cells. We observed that subjects have varying levels of neutralizing antibodies against other SFV-complex members, but this breadth generally did not extend to distantly related VEEV, with the majority of subjects exhibiting VEEV plaque reduction neutralizing titer (PRNT) values below the limit of

**Table 1. Summary of subject data.** Subjects with confirmed or suspected CHIKV infection were enrolled in either an endemic cohort (Ponce, Puerto Rico; color coded in orange), a non-endemic cohort (Portland, Oregon; color coded in blue), or an alphavirus naïve cohort (Portland, Oregon; color coded in black). Subjects are assigned an ID with age, country of birth, country of infection, range of time-post infection for serum collection, CHIKV PRNT$_{50}$ at time of primary blood draw, and reported symptoms displayed.

| Subject ID | Age at time of infection | Country of birth | Country of infection | Range of time post-infection for serum collection (Years) | CHIKV PRNT$_{50}$ | Symptoms |
|---|---|---|---|---|---|---|
| 1 | 45 | Puerto Rico | Puerto Rico | 2.8–6 | 12673 | fever, muscle/joint pain, headache |
| 3 | 13 | Puerto Rico | Puerto Rico | 4.3–5.1 | 8464 | fever, muscle/joint pain, rash |
| 8 | 17 | Puerto Rico | Puerto Rico | 2.8–5.3 | 11834 | fever, muscle/joint pain, rash, malaise |
| 13 | 12 | Puerto Rico | Puerto Rico | 3.4–4 | 59931 | rash |
| 14 | 13 | Puerto Rico | Puerto Rico | 3.4 | 14347 | fever, joint/muscle pain, rash, malaise |
| 16 | 26 | United States | El Salvador | 1.1–6.9 | 17552 | fever. muscle/joint pain, rash, headache, malaise |
| 17 | 48 | United States | Papua New Guinea | 24.3 | 81.8 | fever, headache, malaise |
| 18 | 26 | India | India | 9.3–12.4 | 1202 | fever, muscle/joint pain, malaise |
| 19 | 30 | India | India | 8.7–11.4 | 1130 | fever, muscle/joint pain, rash, headache, malaise |
| 20 | 27 | India | India | 7.9 | 12565 | fever, muscle/joint pain, rash, headache, malaise |
| 21 | 24 | United States | Haiti | 3.5–7.4 | 5996 | fever, muscle/joint pain, rash, headache, malaise |
| 22 | 23 | United States | Haiti | 4–8.2 | 17924 | fever, muscle/joint pain, rash, headache, malaise |
| Naïve 1 | N/A | United States | N/A | N/A | <1:20 | N/A |
| Naïve 2 | N/A | United States | N/A | N/A | <1:20 | N/A |
| Naïve 3 | N/A | United States | N/A | N/A | <1:20 | N/A |

detection. Similarly, interrogation of the MBC compartment following natural infection identified MBCs capable of recognizing both CHIKV and MAYV. Additionally, we looked for the presence of antibodies and MBCs that recognize the E2 B domain, which has previously been implicated as a potential target for broadly cross-neutralizing antibodies [22,24]. The results of this study indicate that natural infection with CHIKV elicits a robust and durable immune response that would ostensibly be protective against repeat infection with CHIKV as well as related Semliki Forest complex alphaviruses for years to decades after initial infection. This cross-reactivity might contribute to the restriction of transmission of closely related alphaviruses in arbovirus endemic regions.

## Results

### Study subjects

Twelve subjects with a confirmed or suspected history of CHIKV infection that occurred between 1992 and 2016 were used for this study (Table 1). Individual subject sera and peripheral blood mononuclear cells (PBMC) were obtained from timepoints ranging from 1–24 years post-infection. Five of the subjects are from a larger endemic cohort of arbovirus immune subjects in Ponce, Puerto Rico (color coded in orange), these infections were PCR confirmed. Seven subjects are from a larger non-endemic (travelers) cohort of arbovirus exposed individuals based in Portland, Oregon (color coded in blue) who were identified through clinical and travel history as well as serology testing (Table 1). Each of these subjects reported the incidence of at least one symptom consistent with alphavirus infection (Table 1).

Based on initial screening, subjects with 50% plaque reduction neutralization titers ($PRNT_{50}$) of >1:20 against CHIKV were presumed to be CHIKV-immune.

## Alphavirus specific neutralization and antigenic relationship by subject

Immune serum from twelve subjects with presumed or confirmed CHIKV infection history and three naïve subjects (Table 1) were used in neutralization assays against a panel of five alphaviruses of the SFV antigenic complex including CHIKV, ONNV, MAYV, UNAV, and RRV, as well as VEEV, which is a representative virus from the VEEV antigenic complex. Amino acid sequences for E1, 6K, and E2 were used to generate the phylogenetic tree (Fig 1A) to demonstrate the genetic relatedness of the viruses used in this study. We conducted 50% plaque reduction neutralization tests ($PRNT_{50}$) for each of the sera against the panel of alphaviruses to determine antigenic breadth and durability following alphavirus infection (Fig 1B, 1C and 1D and S1 Table). Serum samples from five endemic subjects, 4 longitudinal and 1 single time-point, (Fig 1B) and seven non-endemic subjects, 5 longitudinal and 2 single time-points, (Fig 1C) were tested for serum neutralization. All 12 subjects had anti-CHIKV neutralizing antibodies with the highest levels of detection observed for endemic Subject 13 (V2) and non-endemic Subject 16 (V2), which were 4.0 and 6.9 years out from initial infection, respectively, indicating the presence of anti-CHIKV immunity lasting for greater than 20 years following natural infection in both endemic and non-endemic transmission settings (Fig 1B and 1C). Anti-CHIKV neutralizing antibody levels were lowest for non-endemic Subject 17, which demonstrated the highest level of neutralizing antibodies against RRV with a $PRNT_{50}$ of 1120. This result leads us to suspect subject 17, who was infected in Papua New Guinea, may have experienced a primary RRV infection with cross-reactive antibodies against CHIKV (Table 1 and Fig 1C). Interestingly, this person still had durable heterotypic immunity even at >20 years post infection or as an alternative this person may have undergone infection with the same or a related virus. When quantifying cross-neutralizing antibodies against the other five alphaviruses, we found that neutralizing antibody levels were highest for ONNV, which is the closest related of the five viruses to CHIKV; and cross-neutralizing antibody levels decreased the more phylogenetically divergent the virus is from CHIKV (Fig 1D). Not surprisingly, high levels of CHIKV-neutralizing antibodies correlate with higher levels of cross-neutralizing antibodies. Our statistical analysis showed CHIKV neutralizing antibody titers were significantly higher than cross-neutralizing antibody titers against MAYV, UNA, RRV, and VEEV but not ONNV, which is expected given the phylogenetic similarity of CHIKV and ONNV (*** p = 0.0001, **** p = <0.0001) (Fig 1D). Overall, for 9 of the 12 subjects that we serologically profiled at more than one time point post-infection, we found longitudinal changes in both homotypic and heterotypic neutralization over time to be variable compared to the original blood draw but overall antibodies remained stable overtime (Fig 1B and 1C).

We next characterized the antigenic relationship between distinct alphaviruses using antigenic cartography, which has previously been implemented to describe the antigenic relatedness of dengue and influenza viruses [28,29]. Antigenic maps provide an alternate means of using neutralization titers to evaluate antigenic rather than genetic similarities between viruses. We found that CHIKV and ONNV are the most antigenically similar, consistent with the phylogenetic relationship between these two viruses (Fig 1A), suggesting that antibody responses against these viruses share antigenically conserved epitopes; whereas VEEV and RRV are placed at a greater distance from CHIKV, again consistent with the phylogenetic relationships between the viruses (Fig 1A). All sera tested cluster around CHIKV and the closely related ONNV, except for Subject 17, which clusters most closely to RRV (Fig 2A). We next plotted sera and viruses for subjects for who we have serial serum samples, finding that for most

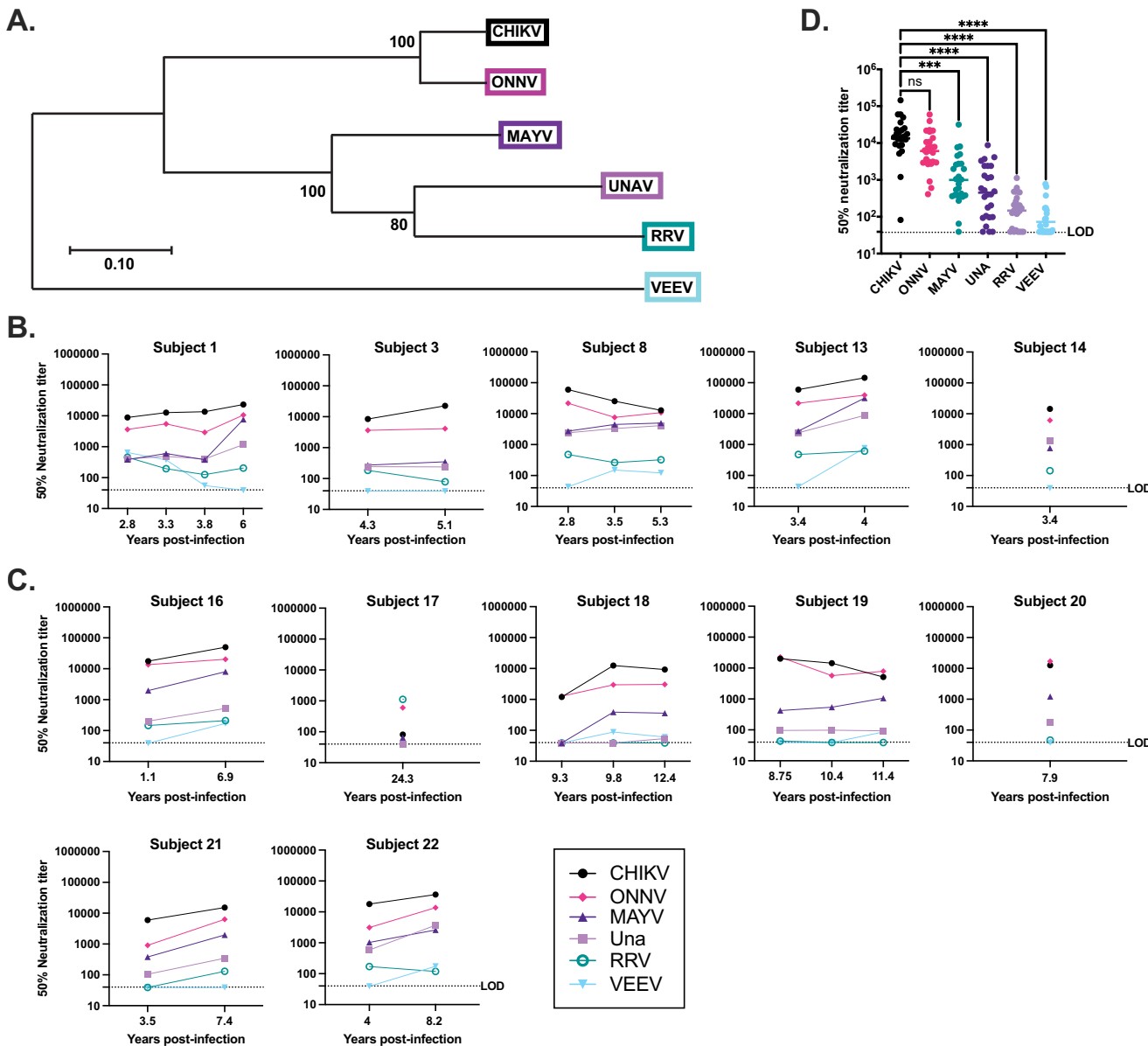

**Fig 1. Longitudinal serology for endemic and non-endemic patients.** (A) Phylogenetic tree produced using the E1, 6k, and E2 amino acid sequences for the six alphaviruses under investigation; viruses are color coded to match serology graphs. (B, C) Sera samples from each subject were tested for neutralization activity against CHIKV, ONNV, RRV, MAYV, Una, and VEEV by plaque reduction neutralization titer assays (PRNT) performed on confluent monolayers of Vero cells. Shown are the average 50% reduction titer values ($PRNT_{50}$) calculated by variable slope non-linear regression using Prism software. Longitudinal serology is shown for 9/12 human subjects. Additional samples for the other human subjects were unavailable. Endemic subject serologic profiles are shown in (B). Serology for non-endemic subjects is shown in (C). (D) Summarizes the breadth of cross-neutralization data for both endemic and non-endemic subjects at all time points presented in (B) and (C). The statistical analysis to compare grouped cross-neutralizing $PRNT_{50}$ values to CHIKV $PRNT_{50}$ was completed using an ANOVA and Friedman's test ***p = 0.0001, ****p = <0.0001. Limit of detection (LOD) is 40, samples below the LOD were assigned an arbitrary value of 39.

subjects, relative antigenic distances between sera and viruses shifted little over time, with the exception of subject 1, who, on their third blood draw, shifted to a position on the map much farther from the other alphaviruses apart from CHIKV. This pattern is consistent with increased virus specificity and narrowed neutralization breadth over time. This finding was either not evident or less prominent for the remaining subject sera. Longitudinal samples (Fig 2B) continue

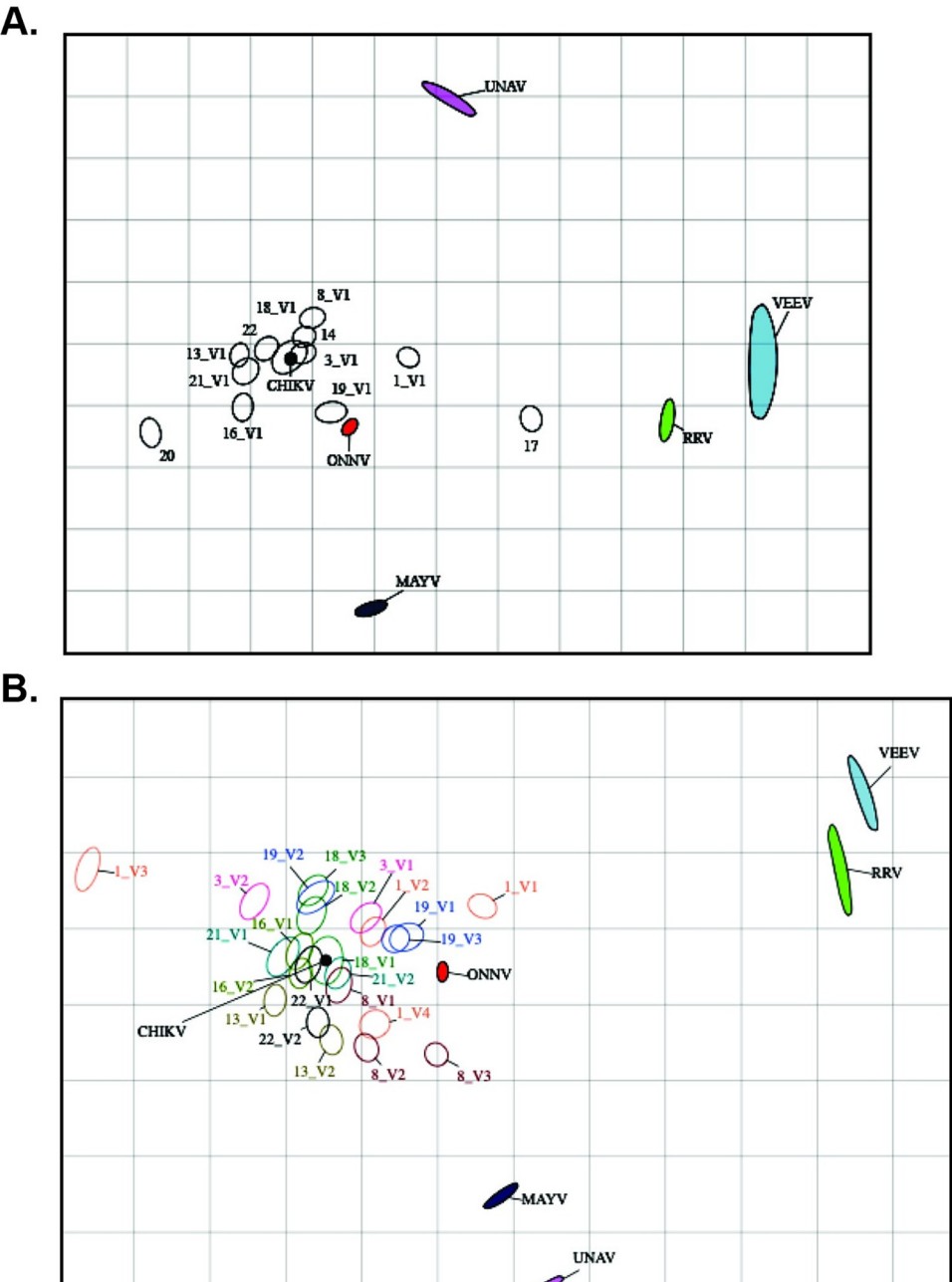

**Fig 2. Antigenic cartography to map human subject alphavirus cross-neutralization by human sera.** Antigenic map shows the relative antigenic relatedness between CHIKV, ONNV, RRV, MAYV, UNAV, and VEEV. Each unit of antigenic distance (AU), the length of one side of a grid square, is equivalent to a two-fold dilution in a neutralization assay. Sera are shown as open ellipses and labeled by subject number. Each virus is shown as a color filled ellipses and is colored according to virus strain (Fig 1A). The size and shape of each ellipse is the confidence area of its position. In making the map, each sera is initially plotted on top of the virus it most potently neutralizes and then pairwise distances between each sera:virus combination are calculated as a fold-difference in titer between the most potently neutralized virus and each other virus. The map is then optimized to place each virus relative to the serum samples in a manner that minimizes error between pairwise fold-differences. The closer a virus is to another virus, the more antigenically related the two are. Sera are initially plotted nearest to the virus they most potently neutralize with subsequently increasing distance to other viruses in descending neutralization potency against each virus. The antigenic map in (A) reflects each human subject at the primary blood draw, and (B) is representative of longitudinal sampling.

to cluster most closely to CHIKV suggesting maintenance of CHIKV-specific and cross-reactive antibodies over time (Fig 2B).

## Dissecting the role of E2 B domain in homotypic and heterotypic neutralization

Conservation of the E2 B domain among members of the SFV complex has been shown to correlate with antibody cross-reactivity [22,30]. The E2 B domain amino acid sequences for CHIKV, ONNV, MAYV, UNAV, and RRV are highly conserved (ranging from 56 to 88% sequence identity) sharing clusters of amino acids distributed across this region of E2, while VEEV shares only 27% sequence identity (Fig 3A and 3B). When viewed in a structural model, the organization of the E1-E2 monomer and arrangement in the spike trimer demonstrates the accessibility of antibody binding to the E2B domain (Fig 3C and 3D). To explore the cross-neutralizing potential of E2 B domain specific antibodies, we first depleted MAYV E2B-specific antibodies by adsorbing subject immune sera against magnetic beads coated with purified MAYV E2 B domain polypeptide (S1A and S1B Fig). Serum samples were incubated with MAYV E2 B domain bound beads, beads alone, or in the absence of beads. Following depletion, sera were evaluated for changes in neutralizing antibody titers against both CHIKV and MAYV relative to controls (Fig 4). Depletion with recombinant MAYV E2 B domain protein did not alter homotypic CHIKV neutralization titers (Fig 5B and 5E), where no significant difference in CHIKV neutralization titer was observed between control and E2 B depletion. However, MAYV neutralization titers significantly decreased compared to control beads (** $p = 0.0045$) for all subjects except for Subject 17, which was excluded from statistical analyses due to the uncertainty of infection history (Fig 5F). Specifically, heterotypic $PRNT_{50}$ titers against MAYV dropped nearly five-fold ($0.208 \pm 0.071$-fold change) whereas control depleted

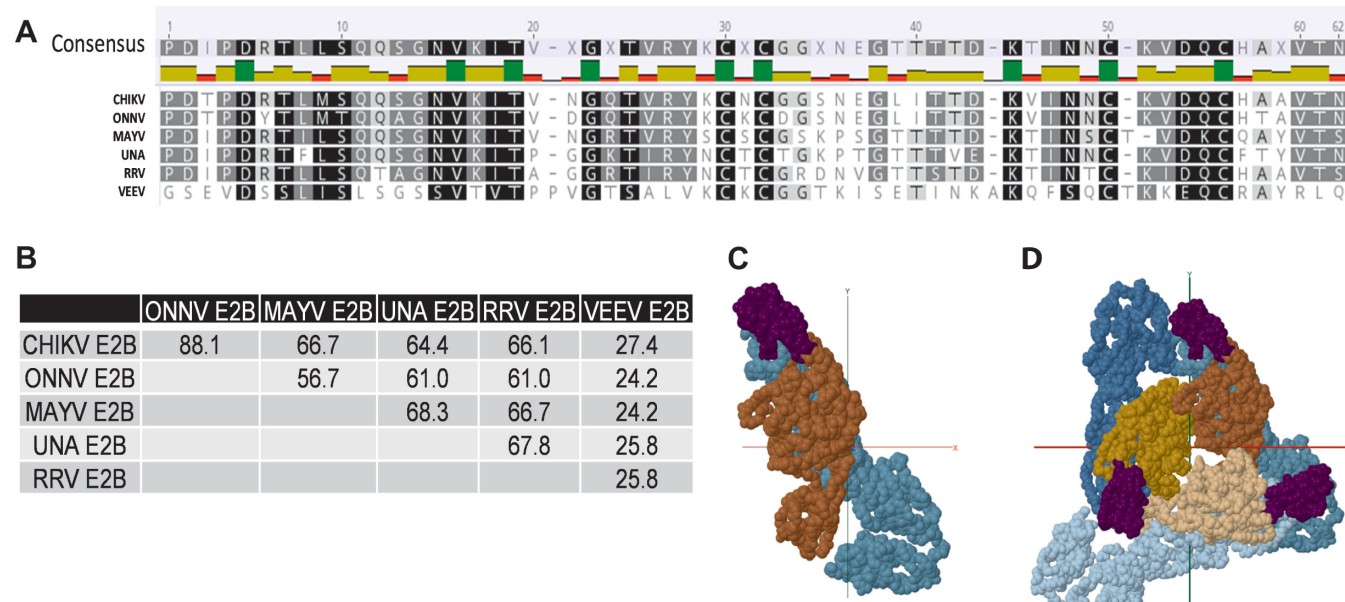

**Fig 3. Comparison of Alphavirus E2 B domains.** (A) Amino acid sequence alignment was performed using Geneious software for the E2 B domains of the alphaviruses examined in this study. Regions of 100% homology are highlighted in black, 80–100% similarity is dark grey, 60–80% similarity is light grey, and less than 60% similarity is in white. (B) Matrix depicts the amino acid sequence identity as a percentage. (C) Top-down view of the organization of the Mayaro Virus E1:E2 monomer (Teal:Brown) shown with the E2 B domain annotated in purple. (D) E1:E2 trimer spike organization depicted with the E2 B domain annotated in purple, E1 in shades of teal, and E2 in shades of brown.

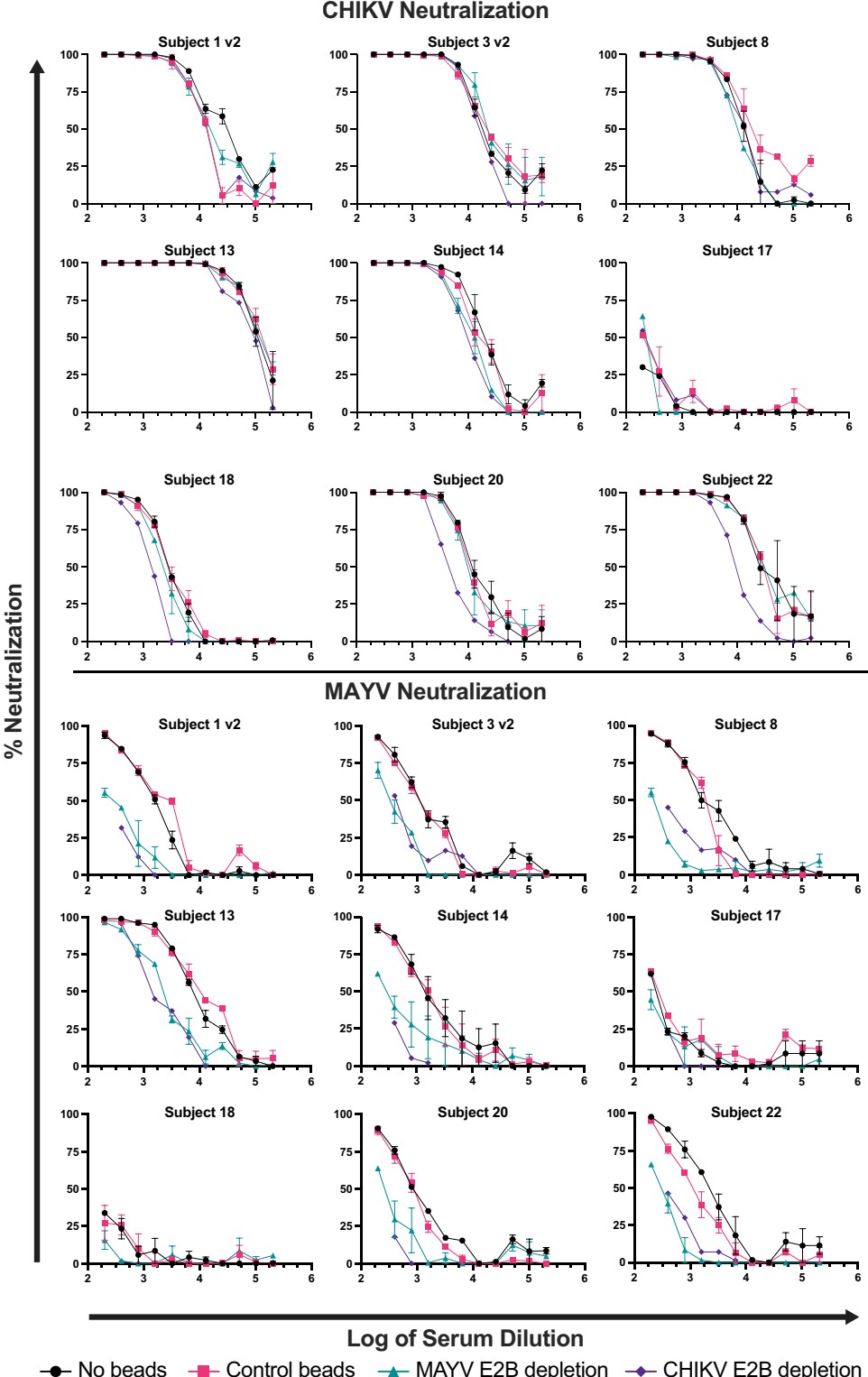

**Fig 4. Impact of depletion of E2B-binding antibodies on CHIKV and MAYV neutralization.** His-tagged CHIKV or MAYV E2 B domain bound to magnetic beads (or control beads alone) was adsorbed by diluted human serum for 4 hours and the beads were pulled off with a magnet. Following depletion, the sera was used in both CHIKV and MAYV neutralization assays. Human sera samples from the first blood draw were diluted 1:2 from 1:100 to 1:102,400. "No beads" is diluted serum only in black, CHIKV E2B absorbed human sera is in purple, MAYV E2B absorbed human

sera is in teal, and control beads bound to diluted human sera is in pink. The data are representative of 3 biological experiments completed with duplicate samples.

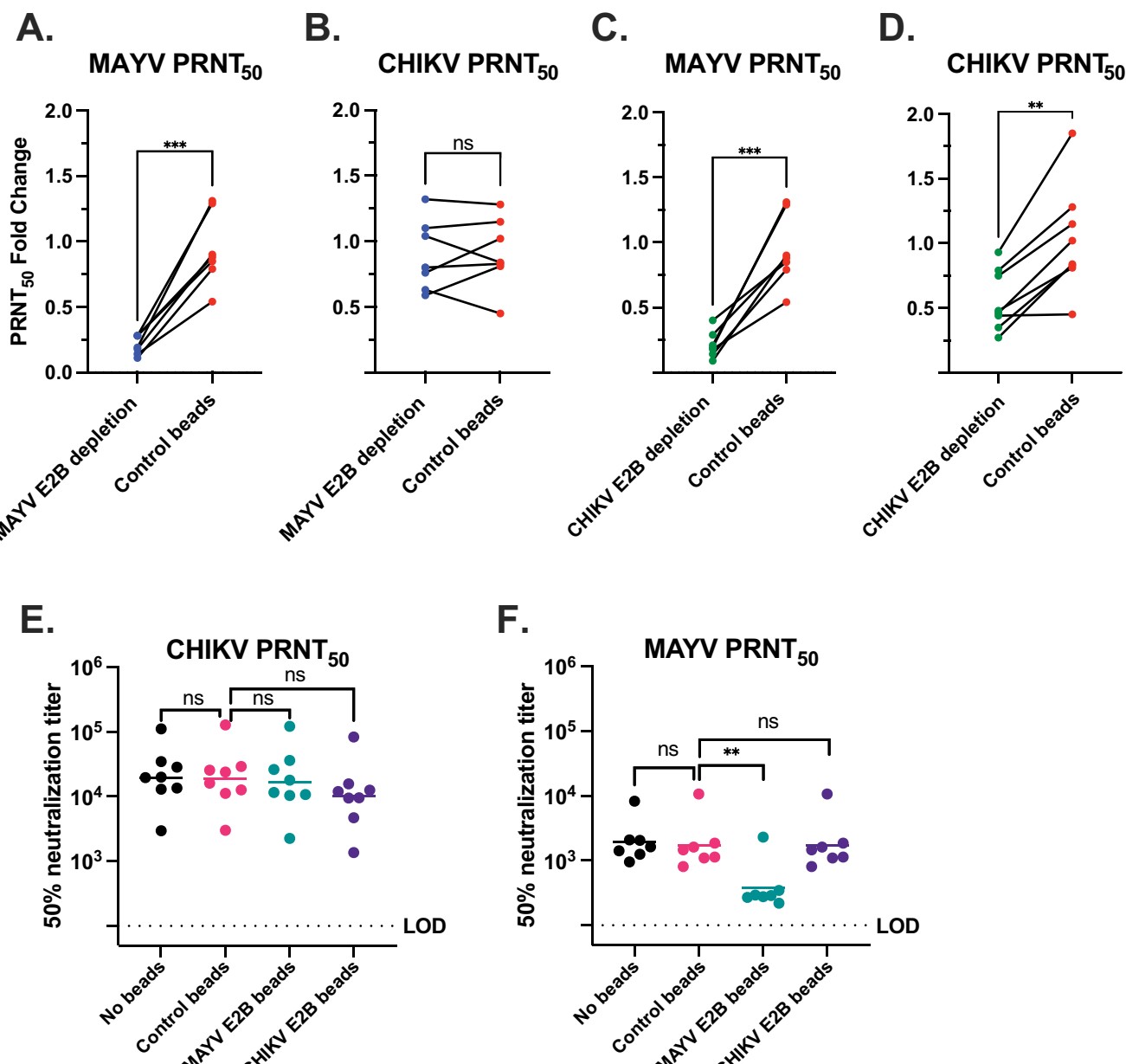

**Fig 5. Analysis of changes in CHIKV and MAYV neutralizing antibody titers following E2 B domain depletion.** Fold change in neutralizing antibody titers (nAb) of subject serum samples following adsorption against E2B domain coated Ni-NTA or control beads was calculated against non-bead-treated serum samples. Depletion of MAYV E2 B domain-specific antibodies and impact on (A) MAYV or (B) CHIKV neutralizing antibody titer fold change compared to serum with control beads. A paired t-test for comparison of fold change heterotypic MAYV neutralization following MAYV E2B depletion yielded a p value *** = 0.0003 and 0.3276 (ns) for homotypic CHIKV neutralization. Depletion of CHIKV E2 B domain-specific antibodies and impact on (C) MAYV or (D) CHIKV neutralizing antibody titer fold change compared to serum with control beads. A paired t-test for comparison of fold change in heterotypic MAYV neutralization following CHIKV E2B depletion yielded a p value *** = 0.0006 and ** 0.0013 for homotypic CHIKV neutralization. Comparison of changes in (E) CHIKV PRNT$_{50}$ or (F) MAYV PRNT$_{50}$ following E2B depletion relative to no beads or control samples. LOD = 100 with values below the limit of detection graphed as 99. Data were analyzed using a one-way ANOVA with the significant comparison in (F) being ** p = 0.0045. Note Subject 17 was excluded from this statistical analysis as the MAYV neutralization in this subject was low, therefore, the impact on E2B depletion was not detectable.

sera $PRNT_{50}$ titers did not change significantly ($0.937 \pm 0.275$-fold change) (S2 Table). Conversely, homotypic CHIKV neutralization assays following MAYV-E2B bead depletion showed no significant impact on neutralizing antibody titers under either condition compared to the non-adsorbed serum ($0.88 \pm 0.251$ and $1.028 \pm 0.415$-fold change, respectively) (Fig 5B and 5E).

To ensure that our MAYV E2B depletion experiment was not resulting in simply the depletion of MAYV-specific antibodies and strengthen our conclusion for a role of E2B-specific antibodies in cross-neutralization, we next depleted CHIKV E2B-specific antibodies from human sera with the same experimental framework as the MAYV depletion experiment (Fig 4). Indeed, we found that depletion of CHIKV E2B-specific antibodies resulted in significant reduction ($0.214 \pm 0.102$ fold change) of MAYV cross-neutralization compared to control depleted sera ($0.937 \pm 0.275$ fold change) (Figs 4, 5C and 5F and S3 Table), while homotypic CHIKV neutralizing antibody titers were only minimally reduced by CHIKV E2B depletion ($0.558 \pm 0.234$ fold change) compared to control depleted sera ($1.02 \pm 0.415$ fold change) (Figs 4, 5D and 5E and S3 Table). These data support that antibodies induced following CHIKV natural infection target epitopes in addition to E2B but underscores that MAYV cross-neutralizing antibodies induced following CHIKV exposure are predominantly mediated by the E2 B domain.

## Homotypic and cross-reactive alphavirus-specific MBC frequency in immune subjects 1 to 24 years post-infection

To further characterize homotypic and cross-specific immune response in CHIKV immune subjects, memory B-cell (MBC) limiting dilution assays were performed. PBMCs were serially diluted in 96 well plates and stimulated to expand and secrete Abs. These Abs were then analyzed for antigen specificity by ELISA using whole CHIKV and MAYV virions as bait. All subjects had CHIKV-specific MBCs, as remotely as 24 years post-infection (Fig 6A). Cross-reactive MAYV-specific MBCs were present in 10/11 (91%) subjects, with only subject 17 falling below the limit of detection (Fig 6B). We next looked for MAYV E2B domain binding MBCs, finding 9 out of 11 (82%) subjects had MBCs encoding E2B cross-reactive Abs as remotely as 8.7 years post-infection (Fig 6C). When grouping the endemic and non-endemic cohorts for geometric mean MBC frequency analysis, the CHIKV MBC frequency was highest at 9.35 per $10^6$ PBMC compared to 2.5 per $10^6$ PBMC for MAYV and 0.892 per $10^6$ PBMC for MAYV E2B MBCs (Fig 6D). The variability of cross-reactive MBCs attributable to E2B varied by subject (Table 2), ranging from 1.7% to 98% of MAYV-binding MBCs. We explored the relationship between MBC frequency and $PRNT_{50}$ titer finding only a very weak correlation (Spearman $R^2 = 0.126$) between the CHIKV-specific MBC frequency and CHIKV $PRNT_{50}$ titer (P value = 0.2862) (S2A Fig). A similar trend was observed for the relationship between MAYV-specific MBC frequency and MAYV $PRNT_{50}$ titer (Spearman $R^2 = 0.318$, P value = 0.0739) (S2B Fig). This indicates that $PRNT_{50}$ titer is not predictive of MBC response, as the two are distinct and independent B-cell populations. Finally, we found that CHIKV and MAYV-MBC frequencies were highly correlated (Spearman $R^2 = 0.747$, P value = 0.0006) with a ratio of CHIKV:MAYV of about 4:1 overall (Table 2 and S3A Fig). MAYV binding MBC frequency was also highly correlated with MAYV E2B MBC frequency (Spearman $R^2 = 0.656$, P value = 0.002) with approximately 1 in 10 MAYV MBCs also being E2B specific (Table 2 and S3B Fig). Overall, we have shown that similar to serum antibody profiles, CHIKV-infection results in a robust and durable MBC response in both endemic and non-endemic transmission settings, with CHIKV-specific and cross-reactive MBCs detected in 91% of subjects out to 24 years post infection.

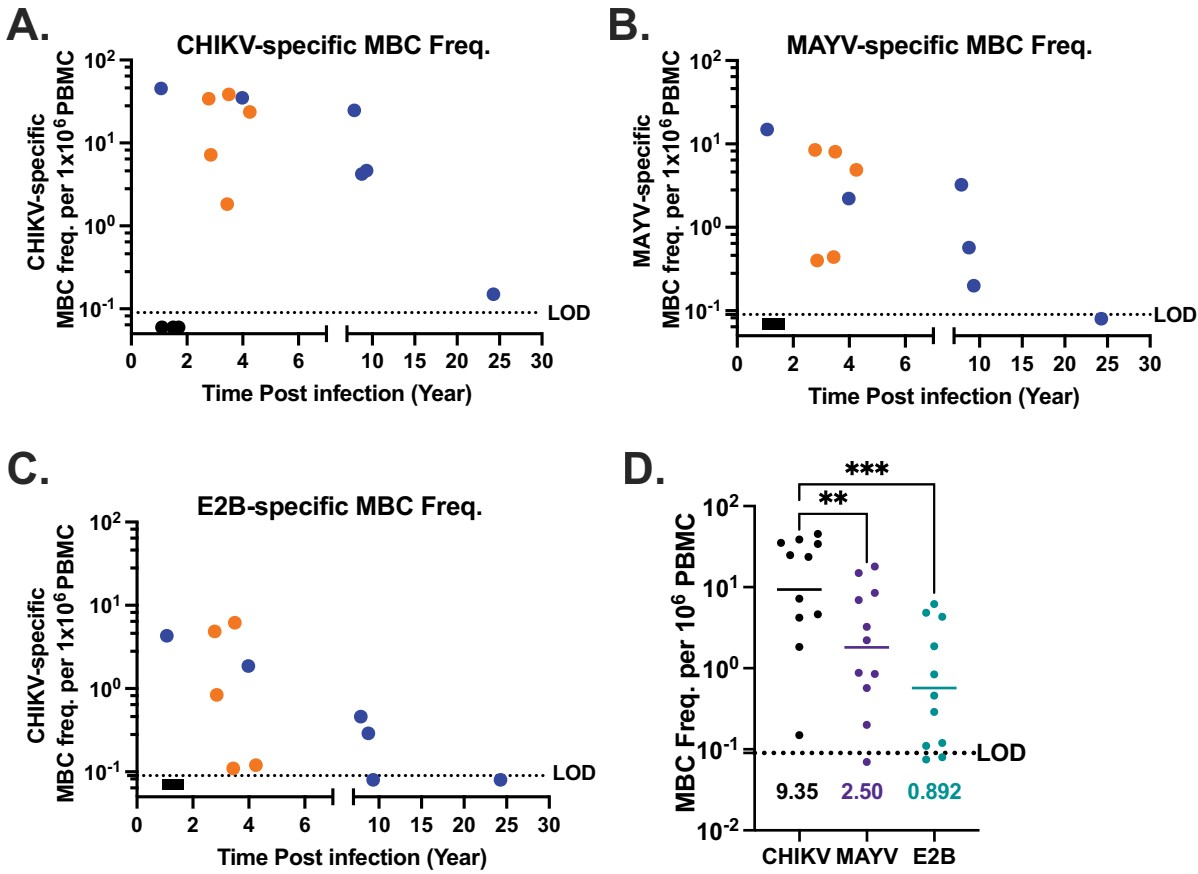

**Fig 6. Antigen-specific MBC frequency per 10^6 PBMC over time in non-endemic cohort (blue n = 6), endemic (orange n = 5), and naïve subjects (black n = 3).** (A) CHIKV-specific MBC frequency as determined by whole CHIKV-ELISA. (B) MAYV-specific MBC frequency determined by whole MAYV-ELISA. (C) E2B-specific MBC frequency determined by MAYV-E2B ELISA. Negative samples and those below the limit of detection were assigned an arbitrary value between 0.05 and 0.09 (LOD = 0.1). (D) Summary of antigen-specific MBC frequency for CHIKV, MAYV, and MAYV E2B with subjects grouped together. Geometric mean frequencies are reported on the graph. P values are the result of a one-way ANOVA [**] p = 0.0044, [***] p = 0.006.

**Table 2. Antigen-specific MBC frequency for non-endemic and endemic cohorts.** Table summarizes subject sampling time post-infection, MBC frequencies for the three antigens tested, and % MAYV MBC attributable to E2B, determined by E2B MBC frequency divided by total MAYV-MBC frequency. ND = not detected, N/A = not applicable.

| | Subject ID | Time Post Infection at primary draw (Years) | CHIKV MBC Freq. | MAYV MBC Freq. | E2B MBC Freq. | % MAYV MBC attributable to E2B |
|---|---|---|---|---|---|---|
| **Endemic** | 1 | 4.5 | 34.15 | 8.50 | 4.85 | 57.0 |
| | 3 | 4.7 | 23.62 | 6.93 | 0.12 | 1.7 |
| | 8 | 4.8 | 10.37 | 0.85 | 0.84 | 98.1 |
| | 13 | 4.7 | 1.83 | 0.88 | 0.11 | 12.2 |
| | 14 | 4.7 | 38.63 | 17.99 | 6.18 | 34.4 |
| **Non-endemic** | 16 | 1.1 | 45.36 | 14.92 | 4.31 | 28.9 |
| | 17 | 24.3 | 0.15 | ND | ND | N/A |
| | 18 | 9.3 | 4.64 | 0.20 | ND | N/A |
| | 19 | 8.7 | 4.22 | 0.57 | 0.29 | 50.0 |
| | 20 | 7.9 | 24.81 | 3.24 | 0.46 | 14.2 |
| | 22 | 4.0 | 35.15 | 2.21 | 1.87 | 84.7 |

## Discussion

Previous characterization of the durability and breadth of CHIKV specific neutralizing antibodies and virus specific MBCs in humans [23,31,32] and mice [33] have been quite limited, but have shown broad serum cross-reactivity. Our data highlight that infection with CHIKV not only elicits durable long-term homotypic neutralizing antibodies years after infection, but also induces neutralizing antibody breadth that extends across multiple SFV complex alphaviruses. Broad neutralization was observed in both endemic and non-endemic subjects with antibody breadth against antigenically distinct viruses remaining stable over time (Fig 1); although we also recognize that the subjects could have been infected, even subclinically, with CHIKV or another alphavirus and this might impact Ab responses to CHIKV or other alphaviruses. Cross-neutralizing antibody responses of a $PRNT_{50}$ of 80 have been suggested to be protective against MAYV [34]. Our data imply that human primary infection with CHIKV has the potential to confer protection against other alphaviruses with the ability to emerge in these same regions. This immunity could reduce patient susceptibility to alphavirus infection and, therefore, has substantial public health relevance as herd immunity could contribute to mitigation of the emergence of closely related arthritogenic alphaviruses. The majority of characterized cross-neutralizing antibodies recognize E2, with many mapping to the linear epitope E2 B domain. There have also been reports of non-neutralizing alphavirus antibodies playing a role in protection, but that was not explored in this investigation [35,36].

We further determined that much of the cross-neutralization and heterologous binding of both LLPC and MBC-derived Abs was attributed to antibodies that recognize the E2 B domain. When serum was depleted of CHIKV or MAYV E2 B domain binding antibodies, cross-neutralization was significantly ablated without significantly reducing neutralization against CHIKV. As such, this further implicates the E2 B domain as an important vaccine antigen for the development of broadly neutralizing alphavirus antibodies and indicates that other antigenic sites are responsible for robust type-specific neutralization. The subjects with the highest percentage of MAYV-specific MBC frequency attributed to the E2 B domain are also the subjects that have the highest fold-change differences in $PRNT_{50}$ following E2B serum depletion. The representation of specific Abs that bind the E2 B domain in the LLPC and MBC compartments varies greatly by subject; however, it is unclear what mechanisms mediate this difference and warrants further investigation.

Further differences were observed between homotypic and heterotypic antibodies in the MBC and LLPC compartments. Geometric mean titers between CHIKV and MAYV differed by 13-fold, compared to differences in MBC frequencies, which differed by less than 4-fold. This difference observed between serum Abs (a product of long-lived plasma cells) and MBC has been shown before in mice and humans where serum Abs are highly specific for the original antigen of infection, while MBCs recognize a greater breadth of antigens, those that are similar but antigenically distinct from the original invading pathogen [37–39]. Our study provides additional evidence of the importance of antibody specificity targeting the E2 B domain following natural CHIKV infection in humans [24,40]. During this investigation we hypothesized that one of our alphavirus immune subjects (Subject 17) had a serum neutralization profile more consistent with prior RRV instead of CHIKV infection. This conclusion is consistent with travel history (infected in Papua New Guinea) and by antigenic cartography (Fig 2A) where the subject antigenically clustered most closely to RRV. This subject was initially identified through a CHIKV-specific neutralization assay screen, and we chose to retain the subject in our analyses because their infection history adds information about the breadth of Ab responses within the SFV-complex and illustrates the importance of specific serological tests to determine infection history. The durability and breadth of the B-cell mediated immune

response to CHIKV indicates that regions with high CHIKV seroprevalence may have a constricted range for closely related alphaviruses as well as point out the importance of specific serologic assays to determine alphavirus infection histories.

## Materials and methods

### Human research ethics

The study has been reviewed and approved by the Oregon Health & Science University Institutional Review Board (IRB#10212) for the non-endemic cohort and Ponce Medical School Foundation Review Board (IRB #180321-VR) for the endemic cohort. Informed written consent was obtained from subjects upon initiation of their participation in the study. Written formal consent for child participants was obtained from the parent/guardian.

### Non-endemic human-cohort population (n = 7)

CHIKV immune individuals in this study were enrolled in a larger study of long-term immunity following infection with the arthropod-borne viruses including DENVs, and ZIKV, as well as for those receiving yellow fever virus (YFV) vaccination. Study subjects with suspected arbovirus infection contacted the long-term immunity study and were offered participation in the study, and following informed consent, provided extensive additional history including other known and suspected arboviral infections, lifetime travel histories, and YFV and Japanese encephalitis virus (JEV) vaccination histories.

### Endemic Human-cohort population (n = 5)

CHIKV immune individuals in this study were enrolled in a larger study of long-term immunity following infection with the arthropod-borne viruses. Study subjects that came to the ER with fever seeking medical attention were approached to enroll in Sentinel Enhanced Dengue Surveillance System (SEDSS). Subjects with PCR confirmed CHIKV infections were offered to participate in the long-term immunity study and following informed consent, provided additional history including other known and suspected arboviral infections, lifetime travel histories, and vaccination histories. Samples were collected, processed and shipped to Oregon Health & Science University for further analysis.

### Sample collection and storage

On enrollment, subjects provided approximately 80 mL of blood, with 30 mL collected in BD serum vacutainers (Becton-Dickson) for serologic studies and stored at -80°C until used for assays. PBMCs were isolated from 50 mL of whole blood collected in BD EDTA or Heparin vacutainers (Becton-Dickson), and stored in liquid nitrogen.

### Viruses

$MAYV_{CH}$ was generated from an infectious clone received from Dr. Thomas E. Morrison (UC-Denver). Mayaro $virus_{BeAr505411}$ (NR-49910); Una $virus_{MAC150}$ (NR-49912); $RRV_{T-48}$ (NR-51457); $ONNV_{UgMP30}$ (NR-51661); and $VEEV_{TC-83}$ (NR-63) were obtained through BEI. $CHIKV_{181/25}$ was generated from infectious clones as previously described [41]. Alphaviruses were grown in C6/36 cells and viral stocks were prepared from clarified supernatants at 72 hours post-infection (hpi) by ultracentrifugation over 10% sucrose (SW32Ti, 70 min at 82,70055 x g). The virus pellets were resuspended in 1X PBS (Corning) and stored at -80°C. Viral limiting dilution plaque assays using Vero cells were performed on 10-fold serial dilutions of virus stocks. The infected cells were rocked continuously in an incubator at 37°C for 2

hours, and then DMEM (Corning) containing 5% FBS (HyClone), 1x Penicillin, Streptomycin, and Glutamine (PSG) (Gibco), 0.3% high viscosity carboxymethyl cellulose (CMC) (Sigma) and 0.3% low viscosity CMC (Sigma) was added to the cells. At 2 dpi, cells were fixed with 3.7% formaldehyde (Fisher) and stained with 0.2% methylene blue (Fisher). Plaques were visualized under a light microscope and counted.

## Neutralization assays—fifty percent plaque reduction neutralization test ($PRNT_{50}$)

$PRNT_{50}$ titers were used to characterize subject sera. Assays were prepared in duplicate (for $CHIKV_{181/25}$). Subject sera were heat-inactivated at 56°C for 30 minutes, then diluted four-fold in MEM supplemented with 2% FBS from a starting dilution of 1:10 for $CHIKV_{181/25}$, 1:20 for assessment against the other viruses ($MAYV_{CH}$, $MAYV_{BeAr505411}$, $Una_{Mac150}$, $RRV_{T-48}$, $ONNV_{UgMP30}$, or $VEEV_{TC-83}$.) 2-fold dilutions were performed in DMEM supplemented with 5% FBS and 1% PSG. Serum dilutions were mixed with an equal volume of 50–100 plaque forming units (PFU) of virus giving a final starting serum dilution of 1:20 for $CHIKV_{181/25}$ and 1:40 for the other viruses evaluated ($MAYV_{CH}$, $MAYV_{BeAr505411}$, $Una_{Mac150}$, $RRV_{T-48}$, $ONNV_{UgMP30}$, or $VEEV_{TC-83}$.). Virus-dilution mixes without sera were prepared simultaneously as controls for input virus PFUs. After incubation at 37°C for 2 hours, virus mixtures were inoculated into individual wells of 24-well plates ($CHIKV_{181/25}$) or 12-well plates seeded with Vero cells, incubated for 2 hours at 37°C 5% $CO_2$, and overlaid with 1% methylcellulose in Opti-MEM (Gibco) supplemented with NEAA, anti-anti, amphotericin B, and 2% FBS ($CHIKV_{181/25}$) or 5% FBS/ DMEM/CMC. Plates were incubated for 2 days ($MAYV_{CH}$, $MAYV_{BeAr505411}$, $Una_{Mac150}$, $RRV_{T48}$, or $VEEV_{TC-83}$) or 3 days ($CHIKV_{181/25}$ and $ONNV_{UgMP30}$) at 37°C and 5% $CO_2$. The overlay was then removed, monolayers were fixed with 80% methanol ($CHIKV_{181/25}$) or 3.7% formaldehyde and stained with 2% crystal violet ($CHIKV_{181/25}$) or 0.2% methylene blue dye, and plaques were enumerated by visual review of each well. Proportion of virus neutralized per well was calculated, and the serum dilution that neutralizes 50% of control input virus ($PRNT_{50}$) was determined by non-linear regression using GraphPad Prism, version 7.0.

## E2 B domain cloning and synthesis

RNA was isolated from the supernatant of $MAYV_{BeAr505411}$ infected C6/36 cells (Quick RNA Viral Kit, Zymo), then purified with RNeasy Mini Kit (Qiagen). cDNA was synthesized using SuperScript IV Reverse Transcriptase (Invitrogen) and the MAYV E2 B domain was amplified with Forward primer: TGAATTCCATATGGTGAGCGGCTGGCGGCTGTTCAAGAAGA TTAGC-CCGGACATTCCGGATAGAAC and Reverse primer: AAGCTTTTAGTGATGG TGATGGTGATGGCTCGTGACGTAAGCCTGACATTTG and cloned into pcDNA3.1. The CHIKV E2 B domain was codon optimized for bacteria with NdeI and HindIII restriction sites, and synthesized by Twist Biosciences: [30] (ATGGGCGTAAGTGGTTGGCGTCTGT TTAAGAAAATCTCGCCGGATACACCAGATCGCACGTTAATGTCCCAACAGTCTG GGAATGTGAAAATTACCGTCAATGGCCAGACTGTTCGCTATAAATGCAACTGTGG AGGTAGCAATGAAGGCCTGATTACGACCGACAAAGTGATCAACAACTGCAAAGTG GATCAGTGTCATGCGGCCGTTACCAACCACCATCACCACCATCATTAA). Both amplicons were cloned into pRSET-B bacterial expression vector with NdeI and HindIII restriction enzymes and transformed into Rosetta (DE3) Competent Cells (Novagen).

## E2 B domain expression and binding to Ni-NTA magnetic beads

Rosetta (DE3) *E.coli* containing the plasmid pRSET-B MAYV or CHIKV E2 B domain were grown in 2X YT broth at 37°C until the $OD_{600}$ reached ~0.6 and then induced with 1 mM final

concentration isopropyl β-D-1-thiogalactopyranoside (IPTG) for 10 hours at 37°C. Cells were pelleted at 10,000 x g for 10 min. Pellets were resuspended in buffer containing 50 mM $NaPO_4^{3-}$ and 300 mM NaCl with 1mg/mL lysozyme and DNase (5ug/mL) pH 8.0 and sonicated for three thirty second cycles at 84W. Cell lysates were centrifuged at 10,000 x g for 10 min, and inclusion body-containing pellets were resuspended with denaturing buffer (8M urea, 30 mM $NaPO_4$, 300 mM NaCl, and 3mM β-mercaptoethanol). Resuspended pellets were rocked for 10 minutes and then incubated at 65°C for 30 minutes. Supernatants were clarified by centrifugation at 416,000 x g for 30 minutes. Supernatant was added to 1 mL of Superflow Ni-NTA resin beads (Qiagen) equilibrated in denaturing buffer. The bead slurry was rocked for 1 hour at RT and then pelleted at 700 x g for 2 minutes. Beads were loaded into a gravity flow column. The beads were washed with 1 mL 20 mM imidazole to remove non-specific binding proteins. The bound protein was eluted with 4 mL of 250 mM imidazole in denaturing buffer, then concentrated to ~750 μL using an Amicon Ultra-15 Centrifugal Filter Unit with 3 kDa cut-off (Amicon), and filtered through a 0.22 μm filter. Filtered elute was loaded onto a Sephacryl S-100HR column that was equilibrated with gel filtration buffer (8M Urea, 100mM Tris pH 8) and separated using an AKTA Start Liquid Chromatograph (GE Lifesciences). Fractions were analyzed for mobility on a NuPAGE 4–12% Bis-Tris gel visualized following staining with Coomassie Brilliant Blue R-250 (Bio-Rad). Fractions containing purified E2 B monomers were combined and then dialyzed in 2-fold steps from 8M urea to PBS using a 3.5K MWCO Slide-A-Lyzer Dialysis Cassette (Pierce). Proteins were quantified using the Nano-Glo HiBiT Lytic Detection System (Promega). Dialyzed fractions were then mixed with 300 μL of PBS equilibrated Ni-NTA Magnetic Beads (Pierce) and rocked overnight at 4°C. Control PBS equilibrated Ni-NTA Magnetic Beads were rocked overnight at 4°C in an equivalent volume of 1X PBS.

## Human serum antibody absorption to Ni-NTA magnetic beads

E2 B domain loaded or control Ni-NTA magnetic beads were washed 3 times with PBS, followed by a blocking wash with DMEM supplemented with 10% human serum (Sigma Human AB serum #H4522). Beads were resuspended homogenously in 2.1 mL of serum-free DMEM and aliquoted evenly between 2 mL centrifuge tubes for each patient and supernatant was removed. Subject serum samples were diluted 1:100 in serum-free DMEM and 1 mL of diluted serum was incubated with E2 B loaded Ni-NTA magnetic beads, control Ni-NTA magnetic beads, or no beads for 4 hours at 4°C. Following incubation, supernatant was removed to new 2 mL tubes and further diluted for use in neutralization assays.

## Neutralization assays with Ni-NTA magnetic bead absorbed human serum

Diluted human serum supernatant following Ni-NTA magnetic bead binding was used in neutralization assays with MAYV$_{BeAr}$ and CHIKV$_{181/25.}$ Serum was diluted 1:2 from 1:100 to 1:102,400 and mixed with media containing 50 PFU of either MAYV$_{BeAr}$ or CHIKV$_{181/25}$. Neutralization assays were then carried out as previously described [42].

## Protein modeling of MAYV structural glycoproteins and alphavirus E2 B domain alignment

MAYV 3D structural model 6W2U, deposited by Powell *et al.*, was downloaded from protein data bank [24,43]. Chains A & E were modeled for Fig 3A, and chains A–C & E–G were modeled for Fig 3B. Chains A and E and A–C & E–G were modeled for monomer and trimer orientations, respectively, using Jmol: an open-source Java viewer for chemical structures in 3D (http://www.jmol.org/). E2 B domain alignment was constructed in Geneious Prime version

11, using the following GenBank accession numbers: CHIKV (SL15649), ONNV (AF079456), MAYV (KT754168), Una (HM147992), RRV (AEC497521), and VEEV (NC001449). Aligned residues were scored using the BLOSUM62 matrix to compare similarity.

## Memory B cell frequency

PBMCs were thawed and resuspended in LDA media (RPMI 1640 medium (Gibco), 1×Antibiotic-Antimycotic (Corning), 1X non-essential amino acids (HyClone), 20 mM HEPES (Thermo Scientific), 50 μM β-ME, and 10% heat-inactivated fetal bovine serum (VWR)). Cells were serially 2-fold diluted (10 wells per cell sample) starting with 3–5 x $10^5$ PBMCs per well at the highest concentration and cultured in in 96-well round-bottom plates in a final volume of 200 μL per well. Cells were stimulated with IL-2 (Prospec) 1000U/mL and R848 (InvivoGen) 2.5μg/mL [44]. To determine background absorbance values, supernatants were used from 8 wells of unstimulated PBMCs only. Plates were incubated at 37˚C and 5% $CO_2$ for 7 days. B cell stimulation and expansion was determined by performing ELISAs detecting total IgG.

MBC precursor frequencies were calculated by the semi-logarithmic plot of the percent of negative cultures versus the cell dose per culture, as previously described [45]. Frequencies were calculated as the reciprocal of the cell dilution at which 37% of the cultures were negative for antigen-specific IgG production. Rows which yielded 0% negative wells were excluded, since this typically resides outside of the linear range of the curve and artificially reduced the MBC precursor frequency. For subjects with low frequency of antigen-specific antibody secreting cells frequency was determined by number of positive wells divided by the total number of IgG positive secreting wells, multiplied by one million, giving a frequency per million PBMCs stimulated.

## Antigen-specific ELISAs

Antigen-specific MBC frequencies were calculated by assaying LDA supernatants by antigen-specific ELISAs [45]. Ninety-six half-well ELISA plates (Greiner Bio-one) were coated with 5 x $10^7$ PFU/mL CHIKV or 1 x $10^7$ PFU/mL MAYV in PBS. Plates were incubated for four days at 4˚C, washed with PBS-T (0.05% Tween) and blocked for 1 hour with 5% milk prepared in PBS-T and then 20 μL of LDA supernatants were added to each well and incubated at RT for 1 hour. Plates were washed 4 times with wash buffer, and 50 μL of 1:3,000 dilution of donkey anti-human IgG-HRP (H + L) (Novusbio, NBP1-73319) detection antibody was added and incubated at RT for 1 hour. Plates were washed 4 times with wash buffer, 50 μL of colorimetric detection reagent containing 0.4mg/mL o-phenylenediamine and 0.01% hydrogen peroxide in 0.05M citrate buffer (pH 5) were added and the reaction was stopped after 20 minutes by the addition of 1M HCl. Optical density (OD) at 492nm was measured using a CLARIOstar ELISA plate reader. LDA wells were scored positive at ODs at least 2-fold above background (unstimulated PBMC wells).

## Antigenic cartography

The CHIKV antigenic map was constructed as previously described [28,29] and implemented using the Acmacs Web Cherry platform (https://acmacs-web.antigenic-cartography.org/). Briefly, antigenic maps are constructed by first generating a table of antigenic distances ($D_{ij}$) between each individual virus (*i*) and serum (*j*) using serum titers for each serum-titer pair ($N_{ij}$). To calculate table distance, the titer against the best neutralized virus for that serum is defined as $b_i$ and the distances for that serum are calculated as $D_{ij} = log_2(b_i)-log(N_{ij})$. For the best neutralized virus for that serum, $N_{ij} = b_i$, and this distance will be equal to 0. For the remaining serum-virus pairs, table distance $D_{ij}$ is equivalent to the fold-difference in titer

between $b_{ij}$ and $N_{ij}$. Euclidean map distance ($d_{ij}$) for each serum-virus pair is found by minimizing the error between the table distance $D_{ij}$ and map distance, $d_{ij}$, using the error function $E = \sum_{ij} e(D_{ij}, d_{ij})$, where $e(D_{ij}, d_{ij}) = (D_{ij} - d_{ij})^2$ when the neutralization titer is above 1:20. For viruses with neutralization titers <1:20, the error was defined as $e(D_{ij}, d_{ij}) = (D_{ij} - 1 - d_{ij})^2 (1/1 + e^{-10(D_{ij} - 1 - d_{ij})})$. To make a map and derive $d_{ij}$ for each serum-virus pair, viruses and sera are assigned random starting coordinates and the error function is minimized using the conjugate gradient optimization method.

## Statistical analysis

Statistics and graphs were created with GraphPad Prism 8. Normalized variable slope non-linear regression using upper and lower limits of 100 and 0, respectively, was used to calculate neutralizing antibody titers. Data from Subject 17 was not included in the analysis represented in Figs 5 and 6D because it is unclear which alphavirus the patient was infected with based upon serology.

## Supporting information

**S1 Fig. MAYV and CHIKV E2 B domain protein detection.** Purified E2 B domain detection by (A, C) SDS-PAGE and (B, D) western blot for HiBit-tagged proteins (~8kDa) to confirm that the MAYV and CHIKV E2B proteins were indeed bound to the His beads before use in subsequent assays. In (A, C), samples were heated to 95˚C for 5 minutes then electrophoresed on a 4–12% Bis-Tris gel for 40min at 160V. Gels in (A, B) were loaded with the same samples to detect MAYV E2B and gels in (C,D) were loaded with the same set of samples to detect CHIKV E2B. Gels in (A, C) were stained using the Coomassie Brilliant Blue Staining Solutions Kit to visualize the proteins and confirm the correct protein sizes of 8 kDa. For western blots in (B,D), the gels were transferred to polyvinylidene fluoride (PVDF) membranes using a semi-dry transfer system and probed for HiBit using a 1:200 dilution of LgBiT, according to a HiBit Blotting System protocol and luminescence was visualized. For (A, B), lane 1 is MAYV E2B protein before His bead binding, lane 2 is control His beads only without protein, lane 3 is unbound protein, and lane 4 is MAYV E2B protein bound to His beads. For (C, D), lane 1 is control His beads only without protein and lane 2 is CHIKV E2B protein bound to His beads.
(EPS)

**S2 Fig. Relationship between CHIKV or MAYV MBC frequency and PRNT$_{50}$.** (A) CHIKV MBC frequency compared to CHIKV neutralization titer at primary blood draw, non-parametric Spearman correlation $R^2 = 0.126$. (B) MAYV MBC frequency compared to MAYV neutralization titer at primary blood draw, non-parametric Spearman correlation $R^2 = 0.318$.
(EPS)

**S3 Fig. Relationship between antigen-specific MBC frequencies.** (A) Relationship between MAYV-MBC frequency and CHIKV-MBC frequency non-parametric Spearman correlation $R^2 = 0.747$. (B) E2B-MBC frequency compared to MAYV-MBC frequency non-parametric Spearman correlation $R^2 = 0.656$.
(EPS)

**S1 Table. Compiled PRNT$_{50}$ values for each subject against the six alphaviruses serologically profiled in this study.** Plaque reduction neutralization titer assays were performed to calculate the 50% neutralization titer against a panel of SFV complex alphaviruses and the encephalitic alphavirus VEEV from the endemic cohort (n = 5) and non-endemic cohort (n = 7) at multiple timepoints. The limit of detection was a 1:40 serum dilution and values not

determined are denoted with ND due to insufficient serum volume.
(DOCX)

**S2 Table. PRNT$_{50}$ values and fold change of MAYV E2 B domain depleted serum samples relative to controls.** PRNT assays were performed on serum samples incubated with beads alone or beads coupled with E2 B domain protein. PRNT$_{50}$ values were calculated for each sample using Prism software. Fold change was calculated in Excel and is relative to the appropriate control ($\Delta$1: Fold change in PRNT$_{50}$ titer following E2B bead treatment relative to non-bead treated serum; $\Delta$2: Fold change in PRNT$_{50}$ titer following control bead treatment relative to non-bead treated serum).
(DOCX)

**S3 Table. PRNT$_{50}$ values and fold change of CHIKV E2 B domain depleted serum samples relative to controls.** PRNT assays were performed on serum samples incubated with beads alone or beads coupled with CHIKV E2 B domain protein. PRNT$_{50}$ values were calculated for each sample using Prism software. Fold change was calculated in Excel and is relative to the appropriate control ($\Delta$1: Fold change in PRNT$_{50}$ titer following E2 B bead treatment relative to non-bead treated serum; $\Delta$2: Fold change in PRNT$_{50}$ titer following control bead treatment relative to non-bead treated serum).
(DOCX)

## Author Contributions

**Conceptualization:** John M. Powers, Nicole N. Haese, William B. Messer, Daniel N. Streblow.

**Data curation:** John M. Powers, Zoe L. Lyski, Whitney C. Weber, William B. Messer, Daniel N. Streblow.

**Formal analysis:** John M. Powers, Zoe L. Lyski, Whitney C. Weber, William B. Messer, Daniel N. Streblow.

**Funding acquisition:** Vanessa Rivera-Amill, William B. Messer, Daniel N. Streblow.

**Investigation:** John M. Powers, Zoe L. Lyski, Whitney C. Weber, Michael Denton, Magdalene M. Streblow, Adam T. Mayo, Nicole N. Haese, Chad D. Nix, Daniel N. Streblow.

**Methodology:** John M. Powers, Zoe L. Lyski, Whitney C. Weber, Michael Denton, Nicole N. Haese, Chad D. Nix, Daniel N. Streblow.

**Project administration:** William B. Messer, Daniel N. Streblow.

**Resources:** Rachel Rodríguez-Santiago, Luisa I. Alvarado, Vanessa Rivera-Amill, William B. Messer, Daniel N. Streblow.

**Supervision:** William B. Messer, Daniel N. Streblow.

**Visualization:** Daniel N. Streblow.

**Writing – original draft:** John M. Powers, Zoe L. Lyski, Whitney C. Weber, Nicole N. Haese, William B. Messer, Daniel N. Streblow.

**Writing – review & editing:** John M. Powers, Zoe L. Lyski, Whitney C. Weber, Nicole N. Haese, Rachel Rodríguez-Santiago, Luisa I. Alvarado, Vanessa Rivera-Amill, William B. Messer, Daniel N. Streblow.

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
