## [Decision Letter · Decision Letter 0]

28 Jul 2022

Dear Dr. Streblow,

Thank you very much for submitting your manuscript "Infection with chikungunya virus confers heterotypic cross-neutralizing antibodies and memory B-cells against other arthritogenic alphaviruses predominantly through the B domain of the E2 glycoprotein" for consideration at PLOS Neglected Tropical Diseases. As with all papers reviewed by the journal, your manuscript was reviewed by members of the editorial board and by several independent reviewers. In light of the reviews (below this email), we would like to invite the resubmission of a significantly-revised version that takes into account the reviewers' comments. 

While several of the comments relate to data interpretation and presentational issues, the authors should carefully the address comments made by reviewer 2 on sample size and patient history (required).

We cannot make any decision about publication until we have seen the revised manuscript and your response to the reviewers' comments. Your revised manuscript is also likely to be sent to reviewers for further evaluation.

Sincerely,

Alain Kohl

Associate Editor

Ann Powers

Deputy Editor

While several of the comments relate to data interpretation and presentational issues, the authors should carefully the address comments made by reviewer 2 on sample size and patient history.

Reviewer's Responses to Questions

**Key Review Criteria Required for Acceptance?**

**Methods**

-Are the objectives of the study clearly articulated with a clear testable hypothesis stated?

-Is the study design appropriate to address the stated objectives?

-Is the population clearly described and appropriate for the hypothesis being tested?

-Is the sample size sufficient to ensure adequate power to address the hypothesis being tested?

-Were correct statistical analysis used to support conclusions?

-Are there concerns about ethical or regulatory requirements being met?

Reviewer #1: The objectives are clearly articulated. While the subject sample is small, the manuscript in my opinion is still admissible.

Reviewer #2: -Are the objectives of the study clearly articulated with a clear testable hypothesis stated? Yes

-Is the study design appropriate to address the stated objectives? No

-Is the population clearly described and appropriate for the hypothesis being tested? No

-Is the sample size sufficient to ensure adequate power to address the hypothesis being tested? No

-Were correct statistical analysis used to support conclusions? Not always

-Are there concerns about ethical or regulatory requirements being met? No

Reviewer #3: In their manuscript, Powers et al characterize the neutralizing antibody response in 12 human subjects with suspected or confirmed history of chikungunya virus infection. Five subjects are from CHIK endemic Puerto Rico while 7 subjects are from a non-endemic cohort based in Oregon. Samples collected at various times post infection, ranging from 1.1 to 24.3 years, were evaluated for neutralizing activity against the Semliki Forest complex viruses CHIKV, ONNV, MAYV, Una, RRV, and the distantly related virus, VEEV.

**Results**

-Does the analysis presented match the analysis plan?

-Are the results clearly and completely presented?

-Are the figures (Tables, Images) of sufficient quality for clarity?

Reviewer #1: Yes, on all counts

Reviewer #2: -Does the analysis presented match the analysis plan? I don't understand this question

-Are the results clearly and completely presented? Not always

-Are the figures (Tables, Images) of sufficient quality for clarity? Could be improved

Reviewer #3: The authors successfully recapitulate that neutralizing cross-reactivity is restricted to members of the same serocomplex and attempt to determine whether that cross-neutralizing capacity is derived from E2 B domain binding antibodies. Below are minor concerns.

Line 161 and Figure 4. Please show the non-linear regression / neutralizing activity curve used to determine the IC50 titer. Showing the IC50 titers or other descriptive readout, such as AUC, would be informative, especially if you perform statistical analyses on those derived data points between groups. I understand you show statistical analyses of the fold changes in figure 5 but distilling the data in figure 4 to a statistical comparison between groups would strengthen the conclusions around figure 4. 

Figure 6. What are the black dots?

Figure 7A. Is this just a different analysis of the same data from figure 1b? If I am understanding correctly, figure 7 is the data used to justify the conclusion that MBCs are responsible for much of the cross-neutralizing antibody responses. If this is correct, it may be better to report fold-change on the y axis and just show both the total nAb and MBC-specific data on one graph. Otherwise the MBC argument is harder to grasp by the reader.

**Conclusions**

-Are the conclusions supported by the data presented?

-Are the limitations of analysis clearly described?

-Do the authors discuss how these data can be helpful to advance our understanding of the topic under study?

-Is public health relevance addressed?

Reviewer #1: Please see my overall comments for details

Reviewer #2: -Are the conclusions supported by the data presented? No

-Are the limitations of analysis clearly described? No

-Do the authors discuss how these data can be helpful to advance our understanding of the topic under study? Yes

-Is public health relevance addressed? Yes

Reviewer #3: Major concern:

Line 207. For the conclusion that CHIKV-E2 B binding antibodies, elicited by CHIKV infection, contribute to cross-neutralization, depletion only of MAYV-specific E2 B binders seems counterintuitive. This begs the question, what would depletion of MAYV-specific non-E2 B binders look like in terms of MAYV neutralization? My opinion is that you would need both data sets to strengthen the conclusion that E2 B binding antibodies contribute more to cross-neutralization than any other epitope. The simplest explanation for this data is that if you deplete any MAYV-specific binders, you will see reduction in MAYV neutralization. This conclusion is strengthened by the observation that depletion of MAYV-specific E2 B binders does not impact CHIKV neutralization, suggesting that other epitopes contribute significantly to neutralization. Alternatively, if you had depleted CHIKV-specific E2 B binding antibodies and demonstrated reduction in MAYV neutralization, then the conclusion would be better supported since you are not depleting MAYV-specific binders. Happy to be wrong on this of course and welcome any rebuttal!

**Editorial and Data Presentation Modifications?**

Reviewer #1: (No Response)

Reviewer #2: Rejection; I would reconsider the paper if there are major changes to the way results are presented and if additional data are added.

Reviewer #3: It would be helpful to the reader if you could re-iterate your overall conclusions for each figure at the end of each results section. A lot of the paper seems to be descriptive without the conclusions statements to apply a lot of the descriptive observations.

**Summary and General Comments**

Reviewer #1: This study describes the analyses of heterotypic cross-neutralizing antibodies in convalescent sera from a small cohort of individuals with suspected or confirmed infection with chikungunya virus (CHIKV) belonging to the alphavirus Semliki Forest antigenic complex. The authors observed the presence of broadly neutralizing antibodies targeting other alphaviruses within this complex. They also present evidence of long-term CHIKV-specific memory B-cells in all subjects analysed. The memory B-cell-derived antibodies bound CHIKV and the related Mayaro virus (MAYV), as well as to the highly conserved domain B of the viral surface E2 glycoprotein. Furthermore, depletion of the E2B-specific antibodies from the chikungunya subject convalescent sera significantly ablated MAYV-specific neutralization without affecting CHIKV neutralization. Thus, the authors experimentally confirm that the E2B domain is a key site for cross-neutralizing antibodies which may potentially confer cross-protection.

Overall, I think the manuscript is technically sound and the data worthy of publication. I have some minor comments below:

1. Lines 141 and 143: There is no Fig 2A. Do the authors’ mean to say Fig. 1A or Figs. 1A and 2?

2. Line 434, legend to Fig 3: There is no 'D' panel shown in the Fig. 3.

3. The antibody depletion data are particularly interesting, but I feel that they could have been made more robust with analyses of the E2B-bead captured antibodies. It should be possible to elute such E2B-specific antibodies from the beads and analyse their anti-MAYV neutralization potency in more detail.

Reviewer #2: I'm uploading these comments as an attachment.

Reviewer #3: (No Response)

PLOS authors have the option to publish the peer review history of their article (what does this mean?). If published, this will include your full peer review and any attached files.

Reviewer #1: No

Reviewer #2: No

Reviewer #3: No
---

## [Decision Letter · Decision Letter 1]

19 Jan 2023

Dear Dr. Streblow,

Thank you very much for submitting your manuscript "Infection with chikungunya virus confers heterotypic cross-neutralizing antibodies and memory B-cells against other arthritogenic alphaviruses predominantly through the B domain of the E2 glycoprotein" for consideration at PLOS Neglected Tropical Diseases. As with all papers reviewed by the journal, your manuscript was reviewed by members of the editorial board and by several independent reviewers. In light of the reviews (below this email), we would like to invite the resubmission of a significantly-revised version that takes into account the reviewers' comments. 

Reviewer 2 still has a number of concerns on data interpretation they would like to see addressed. The authors must address these at revision and include updates in both the text and response letter.

We cannot make any decision about publication until we have seen the revised manuscript and your response to the reviewers' comments. Your revised manuscript is also likely to be sent to reviewers for further evaluation.

Sincerely,

Alain Kohl

Academic Editor

Ann Powers

Section Editor

Reviewer 2 still has a number of concerns on data interpretation they would like to see addressed. The authors should address these at revision.

Reviewer's Responses to Questions

**Key Review Criteria Required for Acceptance?**

**Methods**

-Are the objectives of the study clearly articulated with a clear testable hypothesis stated?

-Is the study design appropriate to address the stated objectives?

-Is the population clearly described and appropriate for the hypothesis being tested?

-Is the sample size sufficient to ensure adequate power to address the hypothesis being tested?

-Were correct statistical analysis used to support conclusions?

-Are there concerns about ethical or regulatory requirements being met?

Reviewer #1: I have no issues of concern

Reviewer #2: (No Response)

Reviewer #3: The authors responded adequately to reviewer concerns

**Results**

-Does the analysis presented match the analysis plan?

-Are the results clearly and completely presented?

-Are the figures (Tables, Images) of sufficient quality for clarity?

Reviewer #1: Yes on all counts

Reviewer #2: (No Response)

Reviewer #3: The authors responded adequately to reviewer concerns

**Conclusions**

-Are the conclusions supported by the data presented?

-Are the limitations of analysis clearly described?

-Do the authors discuss how these data can be helpful to advance our understanding of the topic under study?

-Is public health relevance addressed?

Reviewer #1: Yes on all counts

Reviewer #2: (No Response)

Reviewer #3: The authors responded adequately to reviewer concerns

**Editorial and Data Presentation Modifications?**

Reviewer #1: The revisions presented are acceptable. I have no additional suggestions

Reviewer #2: (No Response)

Reviewer #3: Higher resolution versions of the antigenic cartography images

**Summary and General Comments**

Reviewer #1: I believe the revised manuscript is considerably strengthened with additional experiments and therefore am happy for it to be accepted for publication.

Reviewer #2: (No Response)

Reviewer #3: This substantially-revised manuscript highlights the longevity of anti-CHIKV neutralizing antibody responses in humans, recapitulates that alphavirus neutralizing cross-reactivity is restricted to members of the same serocomplex, and identifies an antigenic epitope that contributes to cross-reactive responses. These findings will be of interest to the alphavirus field.

PLOS authors have the option to publish the peer review history of their article (what does this mean?). If published, this will include your full peer review and any attached files.

Reviewer #1: No

Reviewer #2: No

Reviewer #3: No
---

## [Editor Report · Decision Letter 2]

9 Feb 2023

Dear Dr. Streblow,

We are pleased to inform you that your manuscript 'Infection with chikungunya virus confers heterotypic cross-neutralizing antibodies and memory B-cells against other arthritogenic alphaviruses predominantly through the B domain of the E2 glycoprotein' has been provisionally accepted for publication in PLOS Neglected Tropical Diseases.

Best regards,

Alain Kohl

Academic Editor

Ann Powers

Section Editor

---

## [Editor Report · Acceptance letter]

6 Mar 2023

Dear Dr. Streblow,

We are delighted to inform you that your manuscript, "Infection with chikungunya virus confers heterotypic cross-neutralizing antibodies and memory B-cells against other arthritogenic alphaviruses predominantly through the B domain of the E2 glycoprotein," has been formally accepted for publication in PLOS Neglected Tropical Diseases.

Best regards,

Shaden Kamhawi

co-Editor-in-Chief

Paul Brindley

co-Editor-in-Chief
